# Overall success rate of permanent teeth pulpotomy using ProRoot MTA: A systematic review and meta-analysis of randomized clinical trials

Emmanuel J. N. L. Silva[1,2,3]*, Karem P. Pinto[2], Mahmoud Torabinejad[4], Estefano B. Sarmento[2], Jorge N. R. Martins[5,6,7], Marco A. Versiani[1,8], Gustavo De-Deus[1]

1 Department of Endodontics, Fluminense Federal University, Rio de Janeiro, Brazil, 2 Department of Endodontics, State University of Rio de Janeiro, Rio de Janeiro, Brazil, 3 Department of Endodontics, Grande Rio University, Rio de Janeiro, Brazil, 4 Department of Endodontics, School of Dentistry, Loma Linda University School of Dentistry, Loma Linda, California, United States of America, 5 Department of Endodontics, Faculdade de Medicina Dentária, Universidade de Lisboa, Lisbon, Portugal, 6 Unidade de Investigação em Ciências Orais e Biomédicas (UICOB), Faculdade de Medicina Dentária, Universidade de Lisboa, Lisbon, Portugal, 7 Centro de Estudo de Medicina Dentária Baseada na Evidência (CEMDBE), Faculdade de Medicina Dentária, Universidade de Lisboa, Lisbon, Portugal, 8 Dental Specialty Center, Brazilian Military Police, Minas Gerais, Brazil

* nogueiraemmanuel@hotmail.com

## Abstract

### Introduction

This systematic review and meta-analysis aimed to evaluate the success rate of pulpotomy in permanent teeth using ProRoot MTA.

### Methods

An unrestricted search was carried out in 6 electronic databases, until August 2024. The selection of studies adhered to the PIOS criteria, encompassing only randomized clinical trials that assessed the success rate of pulpotomy in permanent teeth using ProRoot MTA through clinical and radiographic evaluations. Risk of bias was assessed using the RoB-2 tool, and meta-analyses were conducted through RevMan 5.3 and R software. To determine the quality of evidence, the GRADE tool was employed.

### Results

The initial search yielded 971 studies. After removing duplicates, 468 studies underwent initial screening, and 32 studies were considered for eligibility. In the final selection, 26 studies were included, and among these, 14 were categorized as having high risk of bias. The analysis of pulpotomy in permanent teeth using ProRoot MTA revealed an overall success rate of 96%, 90%, and 96% at 6-, 12-, and 24-month follow-up periods, respectively, and an annual failure rate of 8%. Meta-analyses indicated a significantly higher success rate for pulpotomies in teeth with open apex. Upon applying the GRADE assessment, an overall moderate level of evidence was observed.

**Data availability statement:** All relevant data are within the paper and its Supporting Information files.

**Funding:** The author(s) received no specific funding for this work.

## Conclusion

Pulpotomy in permanent teeth using ProRoot MTA yields a success rate exceeding 90%, even up to a 24-month follow-up period. Nonetheless, the certainty of evidence supporting these outcomes is moderate, highlighting the requirement for well-designed randomized clinical trials with extended follow-up durations.

**Registration** This systematic review was registered in the PROSPERO database (registration number CRD42023451466).

## Introduction

Minimally invasive dentistry is founded on the principle of preventing or arresting the disease process in its earliest stages while preserving as much healthy tissue as feasible. In Endodontics, there has been an ongoing search of conservative treatments in permanent teeth affected by deep caries, extensive restorative procedures, or trauma, situations that would otherwise necessitate conventional root canal treatment [1,2]. The expanding range of biocompatible materials developed over the years has highlighted pulpotomy as a compelling alternative for addressing these clinical conditions. This procedure involves the partial or complete removal of inflamed coronal pulp tissue while preserving the remaining healthy pulp, covered by a biomaterial that not only supports tissue repair but also sustains pulp vitality [3,4].

Several randomized clinical trials have demonstrated that pulpotomy results in favorable success rates when addressing exposed pulps in permanent teeth [5–8]. Previous systematic reviews have also consistently reported an average success rate exceeding 90% for pulpotomy in permanent teeth, even in cases involving teeth with closed apices or irreversible pulpitis [9,10]. Thus, given that a recent systematic review and meta-analysis revealed that the success rate of primary root canal therapy ranged from 82% to 92.6% [11], it can be reasonably inferred that, when appropriately indicated, pulpotomy in permanent teeth may be equally effective when compared to pulpectomy followed by root canal therapy.

The success rate of pulpotomy is directly affected by the choice of biomaterial used on the exposed pulp, which should exhibit biocompatibility, possess anti-inflammatory properties, and support tissue formation [1,12]. Hence, calcium hydroxide and mineral trioxide aggregate (MTA) have traditionally served as primary pulp-capping agents in vital pulp therapy procedures. While calcium hydroxide demonstrates antimicrobial activity and the potential to induce hard tissue barrier formation [13], its use is constrained by factors including high solubility, low mechanical resistance [14], lack of adhesion, and inadequate sealing ability [15] due to tunnel defects formed in the mineralized barrier [16]. These limitations have been recognized as the primary factors accounting for the reduced success rates observed in pulpotomies performed with calcium hydroxide compared to MTA [1,10,14]. As a result, MTA has been established as the primary pulp-capping agent for pulpotomy in young permanent teeth [12] due to its multiple advantages, including biocompatibility, reduced microleakage, capacity to stimulate a thicker dentinal bridge with fewer defects, and the ability to release growth factors from dentine [9,14,17,18]. Amongst the various brands of MTA available in the market, ProRoot MTA (Dentsply Tulsa Dental, Rolling Hills Drive Johnson City, TN, USA) stands out as the most extensively researched and studied [19], largely owing to its status as one of the pioneering calcium-silicate-based cements to be introduced to the market. ProRoot MTA is an inorganic bioactive compound composed of dicalcium and tricalcium silicate as well as tricalcium aluminate. This material has not only demonstrated promising physical properties

in laboratory studies [20,21] but has also demonstrated high success rates in pulpotomy procedures conducted on permanent teeth, as evidenced by findings from previous randomized clinical trials [5,22–27].

Notwithstanding previous systematic reviews have consistently revealed a high success rate in permanent teeth pulpotomy when employing MTA [1,3,9,10,12], it is noteworthy that these reviews selectively incorporated studies evaluating permanent teeth with distinct characteristics [1,3,9,12] or those that made comparisons between MTA and specific materials [10], thus not encompassing all the studies examining the success of permanent teeth pulpotomy with MTA. Thus, the purpose of this study was to conduct a systematic review and meta-analysis to assess the overall success rate of pulpotomy in permanent teeth when employing ProRoot MTA.

## Materials and methods

### Review protocol and registration

The systematic review protocol was registered in the PROSPERO database (registration number CRD42023451466) and complies with the Preferred Reporting Items for Systematic Reviews (PRISMA) guidelines [28].

### Focused question

This systematic review centered around the following research question: what is the overall rate of success for pulpotomies performed on permanent teeth with ProRoot MTA?

### Search strategy

A comprehensive search was conducted independently by two reviewers (K.P.P and E.J.N.L.S.) across multiple electronic databases, including PubMed, Cochrane Library, Scopus, Web of Science, Embase, and Science Direct. The search was performed without any filters, language restrictions, until August 2024. The search strategy was formulated by combining Medical Subject Heading (MeSH) terms and relevant text words associated with the research focus including ProRoot MTA (MeSH Term), ProRoot MTA, Pro Root MTA, pro-root MTA, Proroot MTA, proroot MTA, mineral trioxide aggregate (MeSH Term), pulpotomy (MeSH Term), vital pulp therapy, permanent teeth, mature teeth, dentition permanent (MeSH Term), permanent dentition, and secondary dentition. The search terms were merged using the Boolean operators 'AND' and 'OR' to construct the search strategies, which are comprehensively outlined in S1 Table. Subsequently, the screening process encompassed a thorough manual review of the references in the chosen studies, and an additional complementary search was performed on OpenGrey literature database.

The articles initially obtained from the search were imported into Endnote X9 Software (Thomson Reuters, New York, NY, USA) to eliminate duplicates. Two authors (K.P.P and E.J.N.L.S.) independently assessed the titles and abstracts, and relevant studies were examined in full to determine eligibility. If there was any disagreement, a third author (G.D.) made the final decision. In cases where the MTA brand was not specified in the study methodology, the authors were contacted via email to certify that it was ProRoot MTA.

### Inclusion criteria

The inclusion criteria for this systematic review were established following the PIOS framework, which is detailed below, as the Comparison (C) element in PICOS did not apply to this particular review [29, 30]:

- P (Population): mature or immature permanent teeth submitted to pulpotomy due to caries or trauma.
- I (Intervention): partial or full pulpotomy using ProRoot MTA.
- O (Outcome): success rate based on clinical and radiographic evaluations.
- S (Study design): randomized clinical trials.

The inclusion criteria for this systematic review encompassed randomized clinical trials specifically assessing the success rate of pulpotomy on permanent teeth with ProRoot MTA. These trials needed to include both clinical and radiographic assessments, with a minimum follow-up period of 6 months. In the clinical evaluation, a minimum of two of the following symptoms should have been assessed: pain, tenderness to percussion, oedema, swelling, or tooth mobility. In terms of radiographic evaluation, at least two of the following signs should have been considered: widening of the periodontal ligament, presence of a periapical lesion, continuity of root formation, or signs of root resorption.

## Exclusion criteria

Randomized clinical trials with follow-up periods of less than 6 months, non-randomized clinical trials, and articles that deviated from the study's focus, such as *ex vivo* investigations, *in vitro* experiments, observational studies, animal studies, case reports, serial cases, letters to the editor, opinions, and review articles, were excluded.

## Screening protocol

The protocol for selecting and screening scientific papers adhered to a "three-stage assessment" process. In the first stage, the titles and abstracts of the studies were reviewed and categorized as either 'irrelevant' or 'relevant,' based on predefined inclusion and exclusion criteria. In the second stage, the studies initially identified as relevant underwent a thorough examination of their full text and were once again categorized using the same predefined criteria. Finally, in the third stage, all the chosen papers underwent data extraction, quality assessment, and evaluation of the level of evidence.

## Data extraction

The data extraction process was carried out independently by two authors (K.P.P and E.J.N.L.S.) and included the following information: author and publication year, tooth type, sample size for MTA, the status of the roots (mature or immature), reason for pulp exposure, patient age range, pulpal diagnosis, the type of pulpotomy (partial or full), duration of follow-up, success rate, and the annual failure rate (AFR). Any discrepancies were resolved by a third author (G.D.).

## Quality assessment

Two authors (K.P.P and E.J.N.L.S.) independently evaluated the quality of the included studies using the RoB-2 tool developed by the Cochrane Collaboration [31]. This tool is specifically designed for assessing the risk of bias in randomized clinical trials and encompasses evaluation in 5 key domains: the randomization process, adherence to intended interventions, handling of missing outcome data, measurement of outcomes, and selection of reported results. Each domain was classified as having 'low risk', 'some concerns' or 'high risk'. The overall risk of bias of a study was considered 'low risk' if all domains were evaluated as 'low risk', 'some concerns' if one or more domains were evaluated as 'some concern' risk, or 'high

risk' if any domain was classified as 'high risk'. In case of discrepancies, a third author (G.D.) was consulted.

## Meta-analysis

Meta-analyses of proportions were conducted using R software (Version 3.6.3, R Foundation for Statistical Computing, Vienna, Austria) in conjunction with the packages meta, metafor, and weightr. Forest plots were created to assess the overall success rate of pulpotomies in permanent teeth using ProRoot MTA at 6, 12, and 24 months of follow-up, as well as the AFR. The success rate was determined by taking into account the number of successful cases relative to the total cases, and the AFR was computed by considering the cases of failure in relation to the total cases and the duration of follow-up. The estimate of intervention's effect was expressed with 95% confidence intervals (CIs). Chi-square test was used to detect statistical heterogeneity with $p$ value set at < 0.1. The $I^2$ statistic was employed to evaluate the degree of heterogeneity, with values greater than 25%, 50%, and 75% signifying low, moderate, or high levels of heterogeneity, respectively. Fixed-effect models were applied in case of low level of heterogeneity, whereas random-effect models were employed in cases of moderate or high heterogeneity. Publication bias were assessed through both visual means, which involved creating funnel plots, and quantitative analysis, using Egger's regression test with a significance threshold set at $p \leq 0.05$. Furthermore, statistical analyses were conducted using RevMan software (Version 5.3, The Cochrane Collaboration) to compare the success rate across various categories, including the status of the root, patient age, pulpal diagnosis, and the type of pulpotomy, and generated corresponding forest plots.

## Grading of evidence

The level of evidence was evaluated using the Grading of Recommendations, Assessment, Development, and Evaluation (GRADE) methodology, with the assistance of the GRADEpro Guideline Development Tool (McMaster University, Hamilton, ON, Canada) [32]. The evaluations were conducted separately by two authors (K.P.P and E.J.N.L.S.) and a third author (G.D.) was consulted in case of discrepancy, considering the following domains: risk of bias, inconsistency, indirectness, imprecision, and publication bias. Each domain was classified as 'not serious', 'serious' or 'very serious' and the overall certainty of evidence was graded into one of four levels: very low, low, moderate or high.

In evaluating the 'risk of bias' domain, it was considered the eligibility criteria for patient selection, the inclusion of a control group, the measurement of the intervention and outcomes, the control of confounding factors in the study design or statistical analysis, and the adequacy of follow-up [33]. Ratings were categorized as 'not serious' if all the included studies scored more than 3 'no' responses for these parameters, 'serious' if there were 2–3 'no' responses, and 'very serious' if 1–2 'no' responses were present. In the 'inconsistency' domain, it was evaluated whether the studies exhibited consistent effects by taking into account point estimates, their associated confidence intervals, and the criteria for assessing heterogeneity [34]. A rating of 'not serious' was assigned when all the included studies displayed consistent results, 'serious' when some studies presented inconsistent results, and 'very serious' when the majority of the studies exhibited inconsistent outcomes. In the 'indirectness' domain, it was examined differences in the population of interest, the nature of the intervention, and the reported outcome. This assessment aimed to ensure the relevance of the study's patient population to those for whom the intervention is intended and whether the studied outcome holds significant importance for patients [35]. A rating of 'not serious' was applied when all the included studies had more than 3 'no' responses for these parameters, 'serious' when there were 2–3 'no' responses, and 'very serious' when 1–2 'no' responses were noted.

In the 'imprecision' domain, the sample size of the included randomized clinical trials and the 95 CI confidence interval surrounding the estimated effect were assessed. It was classified as 'not serious' when the combined sample size was very large (with an optimal information size of at least 300) and the 95% CI for the effect estimate (OR) did not encompass substantial benefit or harm (OR ranging from 0.75 to 1.25). On the other hand, it was deemed 'serious' if the combined sample size was less than 300 or if the 95% CI for the effect estimate (OR) included significant benefit or harm (OR below 0.75 or above 1.25). It was rated as 'very serious' when the combined sample size was less than 300 and the 95% CI for the effect estimate (OR) indicated substantial benefit or harm (OR below 0.75 or above 1.25) [36]. 'Publication bias' was evaluated by visually inspecting funnel plots and applying the Egger's regression statistical test to detect any signs of funnel plot asymmetry [37].

## Results

### Study selection

The initial search yielded 971 studies (S1 Table, Fig 1). No further studies were identified on OpenGrey. Following the removal of duplicates, 468 studies underwent initial screening based on their titles and abstracts. Out of these, 32 studies were considered eligible and underwent a full-text review (S2 Table). After the full-text review, 6 studies were excluded, leaving 26 studies to be included in the current systematic review [5,7,8,22–27,38–54]. A screening of the references in the selected studies did not reveal any additional studies for inclusion in this systematic review.

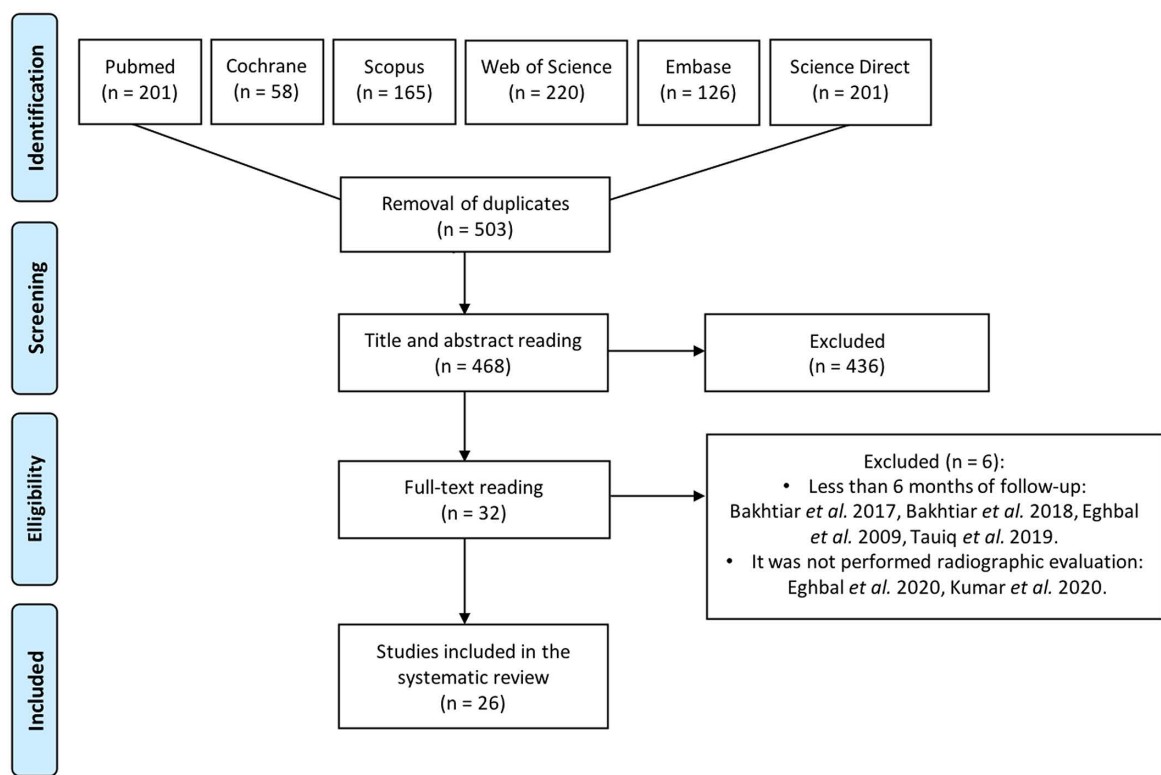

**Fig 1. Prisma flow diagram.**

## Data extraction

Table 1 displays the information collected from the 26 studies included into this systematic review. These studies involved both mature and immature permanent molars with reversible or irreversible pulpitis that underwent either partial or full pulpotomy procedures mainly due to extensive caries lesions. In total, these studies collectively assessed 1118 teeth. It is noteworthy that two articles, Kang et al. [26] and Kang et al. [25], refer to the same clinical trial but with differing follow-up durations. The age range of the patients encompassed a span of 6–82 years, and the duration of the follow-up period ranged from 6 to 78 months. The collective outcomes from the studies indicated a success rate that ranged from 80% to 100%, with an AFR varying between 0% and 15%. The only exceptions were the studies conducted by Kumar et al. [42] which considered teeth with periodontal ligament widening as cases of failure, resulting in a lower radiographic success rate, and the study of Sobh & Ahmed [54] which used apple vinegar or EDTA as final irrigant solution. The other studies used only saline [5,7,22,24,26–28,39–47,51] or sodium hypochlorite [8,23,25,44,48,49,52,53] as irrigant.

## Quality assessment

Fig 2 illustrates the assessment of the risk of bias for the 26 studies included in the analysis. Among these, 14 studies were rated as having a high risk of bias due to deviations from the intended interventions [8,23–27,39,40,44,46,47,49,50,52]. In this domain, factors such as patient adherence to the intervention, participant recalls, and dropout analysis were taken into consideration. The studies with a high risk of bias did not adequately account for potential dropouts in their sample size calculations or failed to assess how dropouts might influence the trial results. On the other hand, the remaining twelve studies were categorized as having a low risk of bias, as no significant biases were detected during their assessment [5,7,22,38,41–43,45,48,51,53,54].

## Meta-analysis

Meta-analyses using fixed-effect models were conducted to assess the overall success rate of pulpotomies in permanent teeth using ProRoot MTA, considering 6-, 12-, and 24-month follow-up periods, as well as the AFR. Fixed-effect models were employed since there was no substantial heterogeneity observed in these meta-analyses, indicated by Chi-squared values ranging from 1.02 to 3.42 and $I^2$ values between 0 and 20%. The success rates for pulpotomy in permanent teeth, assessed at 6-months (11 studies), 12-months (20 studies), and 24-months (6 studies) follow-up periods, were 96% (95% CI: 66–100%; Fig 3A), 90% (95% CI: 72–100%; Fig 3B), and 96% (95% CI: 72–100%; Fig 3C), respectively. The assessment of 23 studies revealed an 8% AFR for pulpotomy in permanent teeth (95% CI: 4–12%; Fig 4), while the funnel plots for all analyses exhibited no substantial asymmetry (S1 Fig). These analyses are corroborated by the results of the Egger's tests, which consistently yielded non-significant p-values (p = 0.3864 for 6-month, p = 0.3124 for 12-month, p = 0.7861 for 24-month, and p = 0.2672 for AFR). These findings provide support for the absence of significant publication bias within the included studies.

Meta-analyses using random-effect models were performed to compare the success rates between pulpotomies based on root status, pulpal diagnosis, and the type of pulpotomy. Random-effect models were employed as heterogeneity could not be confirmed. The analysis revealed a higher success rate for pulpotomy in immature teeth when compared to mature teeth (Odds Ratio: 0.70 [CI: 0.49, 1.00, p = 0.05]; S2a Fig). No significant difference in the success rate was noted between partial and full pulpotomy (Odds Ratio: 1.15 [CI: 0.80, 1.66, p = 0.44]; S2b Fig), or between cases of reversible and irreversible pulpitis (Odds Ratio: 1.68

**Table 1. Data from the included studies.**

| Author and year | Teeth | MTA sample size | Mature or immature teeth | Cause of pulp exposure | Age range (mean age) years | Pulpal diagnosis | Partial or full pulpotomy | Follow-up time: success rate | Annual failure rate |
|---|---|---|---|---|---|---|---|---|---|
| Abueniel et al. 2020 [28] | Central incisors | n=25 | Immature | Trauma | 7.5-9 | Reversible pulpits | Full | 6 months: 100%<br>12 months: 88%<br>18 months: 88% | 8.69% |
| Akhil et al. 2024 [51] | Mandibular molars | n=30 | Mature | Caries | 16-35 (25.03±1.0) | Irreversible pulpitis | Full | 12 months: 88% | 12% |
| Asgary & Eghbal 2013 [5] | Molars | n=208 | Mature | Deep caries | 9-65 (26±9) | Irreversible pulpitis | Full | 12 months: 95% | 5.02% |
| Asgary et al. 2022 [22] | Molars | n=55 | Mature | Deep caries | 14-60 (30.80 ± 1.23) | Irreversible pulpitis | Full | 24 months: 100% | 0% |
| Chailertvanitkul et al. 2014 [23] | First molars | n= 44 | Immature | Caries | 7-10 | Reversible pulpits | Partial | 6 months: 100%<br>12 and 24 months: 100% | 0% |
| Cho et al. 2024 [52] | Premolars and molars | n=44 | Mature | Caries or trauma | 11-82 (44.6± 21.2) | Reversible pulpits | Partial or full | 12 months: 93.9% | 8.1% |
| El Meligy & Avery 2006 [24] | Incisors, premolars and molars | n=15 | Immature | Trauma or caries | 6-12 | – | Full | 6 months: 100%<br>12 months: 100% | 0% |
| Eppa et al. 2018 [39] | – | n=20 | Immature | Caries | 6-14 | – | Full | 3, 6, 9, 12 and 24 months: 100% | 0% |
| Galani et al. 2016 [7] | First and second molars | n=26 | Mature | Caries | 15-36 (20.56 ± 4.38) | – | Full | 18 months: 85% | 10% |
| Kang et al. 2017 [26] | Premolars and molars | n=33 | Mature and immature | Trauma or during caries removal | 6-68 (29.3 ± 14.8) | Reversible pulpits | Partial | 12 months: 96% | 4% |
| Kang et al. 2021 [25] | Premolars and molars | n=33 | Mature and immature | Trauma or during caries removal | 6-68 (29.3 ± 14.8) | Reversible pulpits | Partial | 48-78 months: 90% | NA[*] |
| Keswani et al. 2014 [40] | Molars | n=26 | Immature | Caries | 6-12 (7.87 ± 2.10) | – | Full | 6, 12 and 24 months: 100% | 0% |
| Koli et al. 2020 [41] | Mandibular molars | n=30 | Mature | Caries | 18-35 (24.8 ± 5.95) | Irreversible pulpitis with apical periodontitis | Full | 12 months: 93.3% healed | 6.66% |
| Kumar et al. 2016 [42] | Mandibular molars | n=19 | Mature | Caries | 14-32 (21.20) | Irreversible pulpitis | Full | 6 months: 6.7%<br>12 months: 44.4% | 55.55% |
| Nosrat et al. 2013 [43] | First molars | n=25 | Immature | Caries | 6-10 (8.28 ± 1.27) | Symptomatic and asymptomatic pulpitis | Full | 6 months: 66.0% success (complete apical closure); 34.0% healing (progression of apical closure)<br>12 months: 81.5% success; 18.5% healing | 0% |
| Özgür et al. 2017 [44] | Permanent molars | n=40 | Immature | Caries | 6-13 | – | Partial | 6, 12, 18 and 24 months:<br>Sodium hypochlorite + ProRoot MTA = 94.4%/<br>Saline solution + ProRoot MTA = 100% | 2% |
| Qudeimat et al. 2007 [27] | First molars | n=32 | Mature and immature | Caries | 6.8- 13.3 (10.3 ± 1.8) | – | Partial | 25.4-45.6 months: 93% | NA[*] |

*(Continued)*

**Table 1.** (Continued)

| Author and year | Teeth | MTA sample size | Mature or immature teeth | Cause of pulp exposure | Age range (mean age) years | Pulpal diagnosis | Partial or full pulpotomy | Follow-up time: success rate | Annual failure rate |
|---|---|---|---|---|---|---|---|---|---|
| Ramani et al. 2021 [45] | Mandibular molars | n=101 | Mature | Caries | 18-40 (23.32 ± 4.85) | Irreversible pulpitis | Partial and full | 12 months: Partial: 80.8% Full: 89.8% | 14.85% |
| Singh et al. 2023 [53] | Molars | n=25 | Mature | Caries | 15-45 (29.16 ± 7.63) | Reversible pulpits | Partial | 12 months: 91.3% | 8.33% |
| Sobh & Ahmed 2024 [54] | – | n=40 | Mature | Caries | 18-50 | Irreversible pulpitis | Full | 12 months: Apple vinager as irrigant solution: 78.6%; EDTA as irrigant solution: 57.1% | 17.4% |
| Taha & Khazali 2017 [8] | Molars | n=27 | Mature | Caries | 20-52 (30.3 ± 9.6) | Irreversible pulpitis | Partial | 6 months: 84% 12 months: 83% 24 months: 85% | 4.16% |
| Taha et al. 2022 [46] | Molars | n=50 | Mature | Caries | 10-70 | Reversible or irreversible pulpitis | Full | 6 months: 92.7% 12 months: 91.8% | 8.16% |
| Tozar & Almaz 2020 [47] | Mandibular molars | n=90 | Immature | Caries | 6-15 (8.6 ± 2.2) | – | Partial | 12 months: MTA: 88.8% MTA + laser: 95.5% Overall: 92.2% | 7.78% |
| Uesrichai et al. 2019 [48] | First molars | n=37 | Mature and Immature | Caries | 6-18 (10 ±2.1) | Irreversible pulpitis | Partial | 7-69 months: 92% | NA* |
| Uyar & Alacam 2021 [49] | Molars | n=18 | Immature | Caries | 6-13 (8.0 ±1.4) | Assymptomatic teeth with vital pulps | Partial | 12 months: 94.4% | 5.55% |
| Vu et al. 2020 [50] | – | n=25 | Immature | Caries or trauma | 7–13 (9.2 ± 1.5) | Reversible pulpitis | Partial | 6 months: 95.8% 12 months: 95% | 4.34% |

NA

*: Not available because it was impossible to extract the failures during a specific follow-up time.

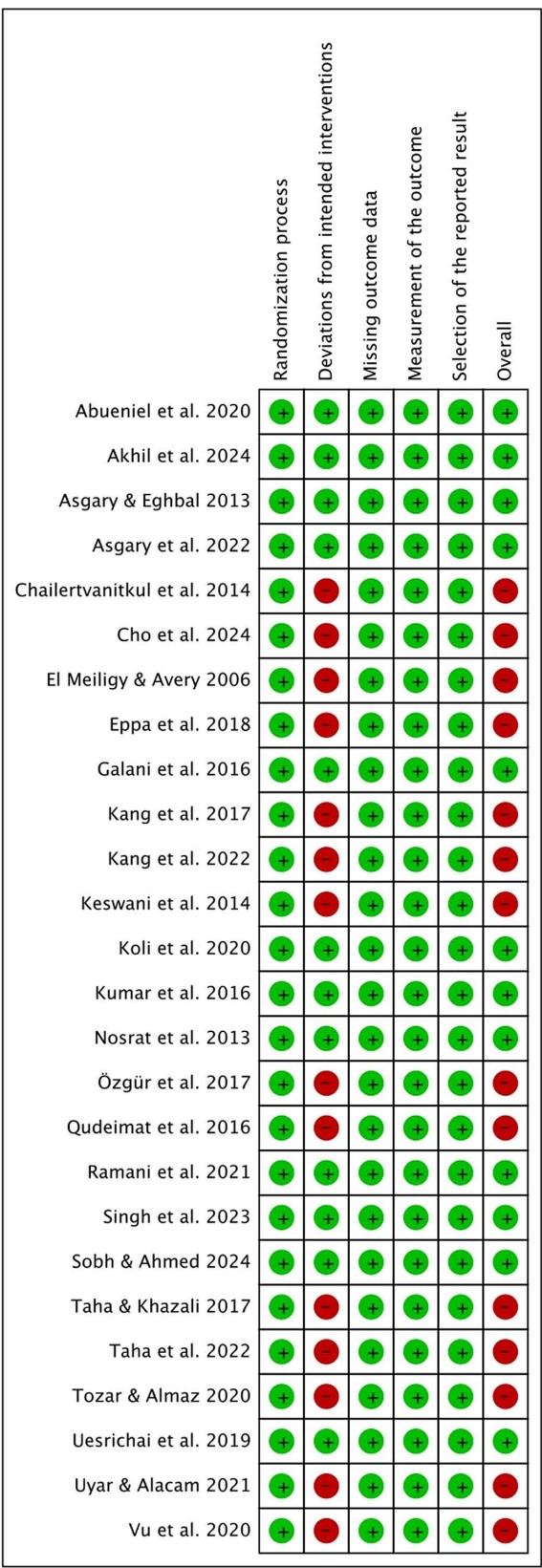

**Fig 2. Risk of bias assessment.**

(A)

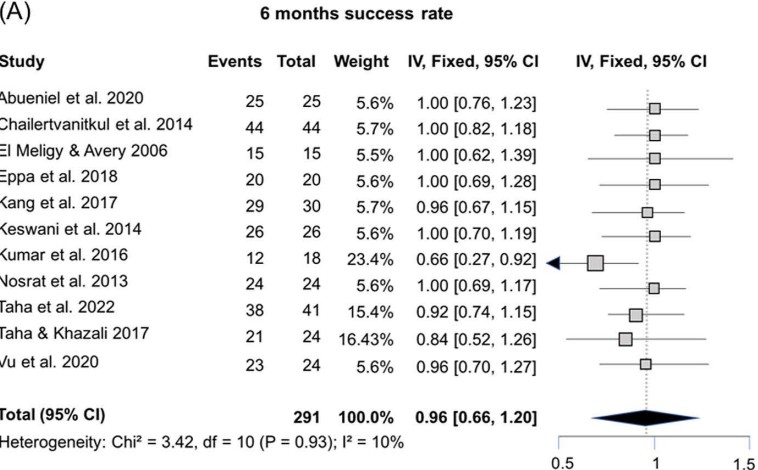

**6 months success rate**

| Study | Events | Total | Weight | IV, Fixed, 95% CI | IV, Fixed, 95% CI |
|---|---|---|---|---|---|
| Abueniel et al. 2020 | 25 | 25 | 5.6% | 1.00 [0.76, 1.23] | |
| Chailertvanitkul et al. 2014 | 44 | 44 | 5.7% | 1.00 [0.82, 1.18] | |
| El Meligy & Avery 2006 | 15 | 15 | 5.5% | 1.00 [0.62, 1.39] | |
| Eppa et al. 2018 | 20 | 20 | 5.6% | 1.00 [0.69, 1.28] | |
| Kang et al. 2017 | 29 | 30 | 5.7% | 0.96 [0.67, 1.15] | |
| Keswani et al. 2014 | 26 | 26 | 5.6% | 1.00 [0.70, 1.19] | |
| Kumar et al. 2016 | 12 | 18 | 23.4% | 0.66 [0.27, 0.92] | |
| Nosrat et al. 2013 | 24 | 24 | 5.6% | 1.00 [0.69, 1.17] | |
| Taha et al. 2022 | 38 | 41 | 15.4% | 0.92 [0.74, 1.15] | |
| Taha & Khazali 2017 | 21 | 24 | 16.43% | 0.84 [0.52, 1.26] | |
| Vu et al. 2020 | 23 | 24 | 5.6% | 0.96 [0.70, 1.27] | |
| **Total (95% CI)** | | 291 | 100.0% | 0.96 [0.66, 1.20] | |

Heterogeneity: Chi² = 3.42, df = 10 (P = 0.93); I² = 10%

(B)

**12 months success rate**

| Study | Events | Total | Weight | IV, Fixed, 95% CI | IV, Fixed, 95% CI |
|---|---|---|---|---|---|
| Abueniel et al. 2020 | 22 | 25 | 4.2% | 0.88 [0.68, 1.03] | |
| Akhil et al. 2024 | 22 | 25 | 3.5% | 0.88 [0.68, 1.04] | |
| Asgary & Eghbal 2013 | 170 | 179 | 17.8% | 0.95 [0.87, 1.07] | |
| Chailertvanitkul et al. 2014 | 41 | 44 | 2.5% | 0.93 [0.83, 1.05] | |
| Cho et al. 2024 | 34 | 37 | 4.5% | 0.94 [0.81, 1.07] | |
| El Meligy & Avery 2006 | 15 | 15 | 1.4% | 1.00 [0.62, 1.39] | |
| Eppa et al. 2018 | 20 | 20 | 1.4% | 1.00 [0.69, 1.28] | |
| Kang et al. 2017 | 24 | 25 | 1.5% | 0.96 [0.70, 1.22] | |
| Keswani et al. 2014 | 26 | 26 | 2.4% | 1.00 [0.70, 1.19] | |
| Koli et al. 2020 | 28 | 30 | 2.8% | 0.93 [0.78, 1.14] | |
| Kumar et al. 2016 | 8 | 18 | 6.7% | 0.44 [0.15, 0.72] | |
| Nosrat et al. 2013 | 24 | 24 | 1.5% | 1.00 [0.69, 1.17] | |
| Ramani et al. 2021 | 86 | 101 | 21.9% | 0.85 [0.79, 0.98] | |
| Singh et al. 2023 | 22 | 24 | 1.5% | 0.91 [0.75, 1.07] | |
| Sobh & Ahmed 2024 | 19 | 23 | 1.4% | 0.68 [0.53, 0.88] | |
| Taha et al. 2022 | 45 | 49 | 5.2% | 0.91 [0.72, 1.18] | |
| Taha & Khazali 2017 | 20 | 24 | 4.5% | 0.83 [0.54, 1.25] | |
| Tozar & Almaz 2020 | 83 | 90 | 12.4% | 0.92 [0.77, 1.09] | |
| Uyar & Alacam 2021 | 17 | 18 | 1.4% | 0.94 [0.71, 1.11] | |
| Vu et al. 2020 | 22 | 23 | 1.5% | 0.95 [0.72, 1.20] | |
| **Total (95% CI)** | | 820 | 100.0% | 0.90 [0.72, 1.19] | |

Heterogeneity: Chi² = 2.10, df = 10 (P = 0.38); I² = 20%

(C)

**24 months success rate**

| Study | Events | Total | Weight | IV, Fixed, 95% CI | IV, Fixed, 95% CI |
|---|---|---|---|---|---|
| Asgary et al. 2022 | 51 | 51 | 14.7% | 0.93 [0.88, 1.05] | |
| Chailertvanitkul et al. 2014 | 41 | 44 | 14.6% | 1.00 [0.83, 1.15] | |
| Eppa et al. 2018 | 20 | 20 | 14.2% | 1.00 [0.69, 1.28] | |
| Keswani et al. 2014 | 26 | 26 | 14.4% | 1.00 [0.70, 1.19] | |
| Özgür et al. 2017 | 39 | 40 | 14.6% | 0.97 [0.74, 1.15] | |
| Taha & Khazali 2017 | 22 | 24 | 27.5% | 0.85 [0.55, 1.21] | |
| **Total (95% CI)** | | 202 | 100.0% | 0.96 [0.72, 1.16] | |

Heterogeneity: Chi² = 1.02, df = 5 (P = 1.00); I² = 0%

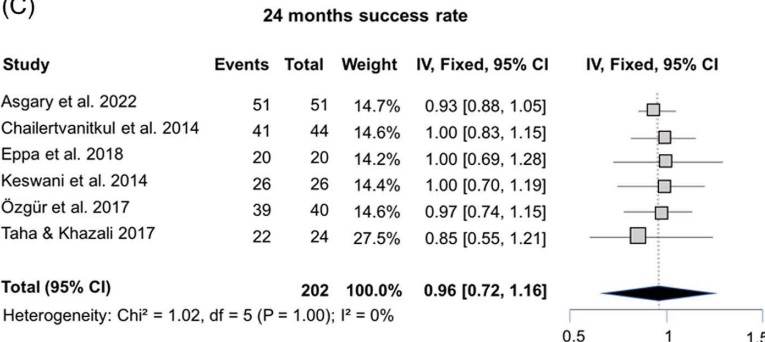

**Fig 3.** (A) **Forest plot of success cases of pulpotomy using ProRoot MTA at the 6-month follow-up showing a 96% of success rate;** (B) **forest plot of success cases of pulpotomy using ProRoot MTA at the 12-month follow-up**

indicating a 90% of success rate; (C) **forest plot of success cases of pulpotomy using ProRoot MTA at the 24-month follow-up showcasing a 96% success rate.**

**Annual Failure Rate**

| Study | Events | Total | Weight | IV, Fixed, 95% CI | IV, Fixed, 95% CI |
|---|---|---|---|---|---|
| Abueniel et al. 2020 | 2 | 23 | 2.8% | 0.08 [0.03, 0.13] | |
| Akhil et al. 2024 | 3 | 25 | 4.0% | 0.12 [0.05, 0.19] | |
| Asgary & Eghbal 2013 | 9 | 179 | 18.9% | 0.05 [0.03, 0.08] | |
| Asgary et al. 2022 | 0 | 51 | 1.5% | 0.00 [0.00, 0.03] | |
| Chailertvanitkul et al. 2014 | 3 | 44 | 1.5% | 0.06 [0.00, 0.05] | |
| Cho et al. 2024 | 3 | 37 | 4.2% | 0.08 [0.04, 0.11] | |
| El Meligy & Avery 2006 | 0 | 15 | 1.4% | 0.00 [0.00, 0.06] | |
| Eppa et al. 2018 | 0 | 20 | 1.5% | 0.00 [0.00, 0.06] | |
| Galani et al. 2016 | 3 | 26 | 2.7% | 0.10 [0.07, 0.13] | |
| Kang et al. 2017 | 1 | 25 | 1.5% | 0.04 [0.01, 0.08] | |
| Keswani et al. 2014 | 0 | 26 | 1.5% | 0.00 [0.00, 0.05] | |
| Koli et al. 2020 | 2 | 30 | 2.9% | 0.06 [0.02, 0.10] | |
| Kumar et al. 2016 | 10 | 18 | 5.4% | 0.55 [0.19, 0.75] | |
| Nosrat et al. 2013 | 0 | 24 | 2.9% | 0.00 [0.00, 0.06] | |
| Özgür et al. 2017 | 1 | 40 | 1.5% | 0.02 [0.00, 0.05] | |
| Ramani et al. 2021 | 15 | 101 | 22.2% | 0.15 [0.11, 0.17] | |
| Singh et al. 2023 | 2 | 24 | 2.8% | 0.08 [0.03, 0.12] | |
| Sobh & Ahmed 2024 | 4 | 23 | 5.1% | 0.17 [0.10, 0.25] | |
| Taha et al. 2022 | 4 | 49 | 1.5% | 0.08 [0.05, 0.12] | |
| Taha & Khazali 2017 | 1 | 24 | 2.8% | 0.04 [0.01, 0.07] | |
| Tozar & Almaz 2020 | 7 | 90 | 7.1% | 0.07 [0.05, 0.09] | |
| Uyar & Alacam 2021 | 1 | 18 | 1.4% | 0.05 [0.02, 0.08] | |
| Vu et al. 2020 | 1 | 23 | 2.9% | 0.04 [0.01, 0.07] | |
| **Total (95% CI)** | | **932** | **100.0%** | **0.08 [0.04, 0.12]** | |

Heterogeneity: Chi² = 3.55, df = 36 (P = 0.40); I² = 20%

0   0.05   0.1   0.15   0.2

**Fig 4. Forest plot showing an annual failure rate of 8% for pulpotomies using ProRoot MTA.**

[CI: 0.97, 2.90, p = 0.06]; S2c Fig). The impact of the underlying cause for pulp exposure on the success rate of pulpotomies could not be statistically assessed due to the limited data available. Only one of the included studies specifically investigated teeth that underwent pulpotomy as a result of trauma [38].

## Grading of evidence

The evaluation of the evidence quality for the included studies was conducted using the GRADE tool and resulted in an overall moderate quality assessment (Table 2). These studies were labelled as having a 'serious' risk of bias because a majority of them failed to adequately address confounding factors, both in their study design and statistical analysis [33]. However,

**Table 2. Assessment of certainty of evidence.**

| Certainty assessment | | | | | | |
|---|---|---|---|---|---|---|
| Participants (studies) | Risk of bias | Inconsistency | Indirectness | Imprecision | Publication bias | Overall certainty of evidence |
| 1118 (26 randomized clinical trials) | Serious[a] | Not serious[b] | Not serious[c] | Serious[d] | None[e] | ⊕⊕⊕○ MODERATE |

a.Most studies showed absence of confounding factors control in the design or statistical analysis.

b.All studies showed consistent results and was not observed unexplained heterogeneity.

c.Populations were representative of the patients for whom the interventions are recommended and patient-important outcomes were assessed.

d.In the meta-analyses of 6-months and 24-months success rate, the pooled sample size was lower than optimal information size, although the 95% CI of the estimative of effect included appreciable benefit or harm.

e.No publication bias was detected in funnel plots and Egger's tests.

the studies received a 'not serious' rating in terms of 'inconsistency' because all of them exhibited consistent results, and unexplained heterogeneity was not observed [34]. The domain 'indirectness' was also classified as 'not serious' since the included studies did not involve indirect comparisons or present indirect results [35]. The populations under study were representative of the patients for whom the interventions are typically recommended, and the assessment of outcomes that matter to patients was ensured.[35] The domain of 'imprecision' was rated as 'serious' because the required optimal information size was not achieved in the 6-month and 24-month meta-analyses, despite the 95% CI for the effect estimate encompassing noticeable benefits or harms [36]. Furthermore, both funnel plots and Egger's test provided evidence of the absence of significant publication bias within the included studies [37].

## Discussion

Vital pulp therapy is a conservative strategy in the field of Endodontics directed at maintaining the well-being and functionality of the dental pulp. This approach encompasses various procedures, including direct pulp capping, partial pulpotomy, and complete pulpotomy, all designed to safeguard the vitality and normal functions of dental pulps that may be exposed due to factors like caries, trauma, or restorative interventions [55]. The aim of this systematic review and meta-analysis of randomized clinical trials was to assess the overall success rate of pulpotomy in permanent teeth when ProRoot MTA was used as the pulp-capping agent. Our results revealed a success rate exceeding 90%, with a notable 96% success rate at the 24-month follow-up period, accompanied by an AFR of 7%. According to several authors, one significant factor contributing to the failure of vital pulp therapy is the potential for bacterial leakage toward the remaining vital pulp in the root canals, often occurring through a defective restoration, thereby compromising the integrity of the pulp complex sealing [9,52–55]. Consequently, it is worth emphasizing that the efficacy of pulpotomy relies heavily on achieving an effective sealing through the use of suitable capping materials and ensuring a proper final restoration. As reported by Alqaderi et al. [9], the success rate of vital pulp therapy may decline over time, underscoring the importance of regular follow-up visits. These visits are essential for the evaluation and potential repair of any flawed restorations, thereby safeguarding the integrity of the dental pulp and enhancing the long-term success of the therapy.

In this review, it was observed that permanent teeth of younger patients with an open apex displayed notably higher success rates in comparison to mature teeth. This outcome aligns with expectations, as the preserved pulp tissue, especially in young and immature teeth, exhibits a high cell count and enhanced vascularity which heightened its resistance against infection and contamination [12]. Furthermore, the presence of a larger population of active dental stem cells within the pulp of immature teeth highlights its significant potential for

regenerative treatments, as these specialized cells possess the remarkable capacity to transform into diverse cell types, thereby contributing to a favorable outcome in pulpotomy [57, 58]. Nonetheless, despite the well-established understanding of the age-related decline in dental stem cell regenerative potential [59], it has also been noted that age itself might not serve as a significant risk factor for the success of pulpotomy in mature teeth [9]. Kunert et al. [53] conducted a long-term retrospective study (1–29 years), assessing patients aged 8–79 who had undergone pulpotomy procedures, and their findings led to the conclusion that the patient's age at the time of pulpotomy did not have a significant impact on the success rates. Moreover, other studies that included patients up to 50 years old have also reported consistently high success rates, [56,60–62] suggesting that vital pulp therapy can be equally effective in elderly patients as it is in younger individuals.

The objective of partial pulpotomy is to eliminate the coronally inflamed pulp while retaining the unaffected deeper pulp, and its advantages compared to full pulpotomy encompass preserving cell-rich coronal pulp tissue for enhanced healing potential, as well as facilitating the ongoing deposition of dentine in the cervical region to keep the structural integrity of the tooth [63]. Therefore, the choice between a partial or full pulpotomy is challenging as it hinges on the accurate clinical identification of non-inflamed pulp tissue [1]. The current analyses, in agreement with previous studies [3,45,64], revealed that, despite inconsistencies in the pretreatment diagnostic criteria among selected studies, there were no statistically significant variations in treatment outcomes when comparing different types of pulpotomy (partial and full) or pulpotomy in teeth with reversible or irreversible pulpitis. These findings carry significance as diagnosing pulp vitality in cases of reversible or irreversible pulpitis remains a clinical challenge, primarily due to the lack of precise diagnostic tools and definitive literature-based criteria to distinguish between these conditions [9,13]. The consistent success rates of pulpotomies in teeth with both reversible and irreversible pulpitis may be attributed to the fact that damage and inflammation in these cases are frequently confined to a portion of the coronal pulp [65]. Furthermore, the anti-inflammatory characteristics of tricalcium silicate materials can facilitate the resolution of remaining inflammation and support the preservation of a healthy pulp tissue [17]. Taken together, these observations suggest that removing the coronally inflamed pulp might be adequate to preserve the vitality and health of the non-affected radicular pulp, making pulpotomy an effective procedure for both emergency and definitive treatment in specific patient populations.

In the current review, the quality of evidence in the included studies was generally moderate. Consequently, while the evidence is trustworthy, future research could potentially exert a substantial influence on the estimate of effect [66]. The majority of the studies included in this review were categorized as high risk in terms of bias, primarily because they did not adequately address patient dropout rates during the follow-up period [8,23–27,39,40,44,46,47,49,50,52]. It was also noted that substantial differences existed in their methodological approaches, pointing to a lack of consistency in the criteria applied for clinical and radiographic analyses, as well as variations in follow-up periods. Additionally, the optimal information size required was not met in the 6-month and 24-month meta-analyses. It is noteworthy to highlight that, in the present study, while the 12-month follow-up demonstrated a 90% success rate, the 24-month follow-up revealed an even higher success rate of 96%. This apparent improvement in success over time could be attributed to the limited number of studies included in the 24-month analysis. On the other hand, this systematic review and meta-analysis, classified as Level 1a (Systematic review of Randomized Controlled Trials) [67], exhibits several strengths, including: (i) a comprehensive literature search across six major databases, reference lists, and grey literature, without any restrictions, encompassing a meticulous assessment of risk of bias and quality of

evidence, carried out by two independent authors, (ii) a detailed meta-analyses to evaluate the overall success rate at three distinct follow-up intervals and to assess the AFR, (iii) a multiple subgroup meta-analyses to evaluate the impact of root status, patient age, pulpotomy type, and pulpal status on the overall success rate of pulpotomy, and (iv) a robust inclusion and exclusion criteria that were used to maintain focus within the review and minimize potential bias arising from study selection.

As observed in this review and consistently emphasized by multiple authors [13,14,68], the current available evidence is inadequate for making definitive conclusions concerning the impact of various factors on pulpotomy outcomes, underscoring the necessity for further high-quality observational studies. A limitation of this review is its potential overlap with previous systematic reviews [10,69] addressing specific aspects of pulpotomy; however, our study uniquely consolidates randomized clinical trial data focusing exclusively on ProRoot MTA's overall success rate in permanent teeth. These future studies should encompass extended follow-up periods, preferably with a minimum of 2 years [70]. They should also implement a more comprehensive methodology, encompassing robust randomization and blinding techniques [12], while striving for uniformity in the criteria used to assess clinical and radiographic outcomes [13], Furthermore, further research should consistently document patient and tooth-specific variables like age, gender, and tooth type, as well as operator-related factors such as the specialty and experience of the investigator, and technical details concerning the type of pulpotomy, hemostatic agents, biomaterial selection, and permanent restorations [14]. To enhance the applicability and generalizability of results, data from recall appointments should also be included [14]. Additionally, it is imperative to broaden the focus beyond clinical effectiveness and incorporate assessments of cost-effectiveness and quality of life in these future investigations [3]. By addressing these aspects, future studies in the field of pulpotomy can contribute to a more comprehensive understanding and ultimately lead to improved outcomes for patients.

## Conclusions

This systematic review and meta-analysis revealed a success rate exceeding 90% for pulpotomy in permanent teeth when employing ProRoot MTA, even after a 24-month follow-up period. While this high success rate offers promise for clinical application, it is noteworthy that the certainty of evidence supporting these results is moderate. Therefore, there is a demand for more extensive, well-designed randomized clinical trials with extended follow-up durations to provide a more comprehensive evaluation of pulpotomy outcomes.

## Supporting information

**S1 Table. Search terms.**
(DOCX)

**S2 Table. Description of the excluded records and reasons.**
(XLS)

**S1 Fig.** Funnel plots for the meta-analyses: (a) success rate at 6-month follow-up; (b) Success rate at 12-month follow-up; (c) success rate at 24-month follow-up; (d) Annual Failure Rate.
(PNG)

**S2 Fig.** The forest plots indicated a statistically significant difference between (a) mature and immature teeth. In contrast, there was no significant difference observed in the comparisons of (b) partial and full pulpotomy and (c) teeth with reversible and irreversible pulpitis.
(PNG)

**S1 File. PRISMA checklist.**
(DOCX)

## Author contributions

**Conceptualization:** Emmanuel Silva, Karem P. Pinto, Gustavo De-Deus.

**Data curation:** Emmanuel Silva, Karem P. Pinto, Marco A. Versiani.

**Formal analysis:** Emmanuel Silva, Karem P. Pinto, Marco A. Versiani, Gustavo De-Deus.

**Investigation:** Emmanuel Silva, Karem P. Pinto, Estefano B Sarmento, Jorge N. R. Martins.

**Methodology:** Emmanuel Silva, Karem P. Pinto, Estefano B Sarmento, Jorge N. R. Martins.

**Supervision:** Emmanuel Silva.

**Validation:** Emmanuel Silva.

**Visualization:** Emmanuel Silva.

**Writing – original draft:** Emmanuel Silva, Karem P. Pinto, Estefano B Sarmento, Jorge N. R. Martins, Marco A. Versiani.

**Writing – review & editing:** Emmanuel Silva, Karem P. Pinto, Mahmoud Torabinejad, Marco A. Versiani, Gustavo De-Deus.

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
