## [Decision Letter · Decision Letter 0]

3 Jan 2025

PONE-D-24-38286Overall success rate of permanent teeth pulpotomy using ProRoot MTA: a systematic review and meta-analysis of randomized clinical trialsPLOS ONE

Dear Dr. Silva,

Thank you for submitting your manuscript to PLOS ONE. After careful consideration, we feel that it has merit but does not fully meet PLOS ONE’s publication criteria as it currently stands. Therefore, we invite you to submit a revised version of the manuscript that addresses the points raised during the review process.

Dear Authors,

Kindly read all the comments given by the reviewers carefully and address them; make the changes in the revised manuscript accordingly.

Best regards and keep well

We look forward to receiving your revised manuscript.

Kind regards,

Mohmed Isaqali Karobari, BDS, MScD.Endo, Ph.D. Endo, FDS, FPFA, MFDS

Academic Editor

PLOS ONE

2. Peer review at PLOS ONE is not double-blinded (https://journals.plos.org/plosone/s/editorial-and-peer-review-process). For this reason, authors should include in the revised manuscript all the information removed for blind review.

3. As required by our policy on Data Availability, please ensure your manuscript or supplementary information includes the following:

Additional Editor Comments:

Dear Authors,

Kindly read all the comments given by the reviewers carefully and address them; make the changes in the revised manuscript accordingly.

Best regards and keep well

Reviewers' comments:

Reviewer's Responses to Questions

**Comments to the Author**

1. Is the manuscript technically sound, and do the data support the conclusions?

Reviewer #1: Yes

Reviewer #2: Yes

Reviewer #3: Yes

Reviewer #4: No

2. Has the statistical analysis been performed appropriately and rigorously?

Reviewer #1: No

Reviewer #2: Yes

Reviewer #3: Yes

Reviewer #4: Yes

3. Have the authors made all data underlying the findings in their manuscript fully available?

Reviewer #1: Yes

Reviewer #2: Yes

Reviewer #3: Yes

Reviewer #4: Yes

4. Is the manuscript presented in an intelligible fashion and written in standard English?

Reviewer #1: Yes

Reviewer #2: Yes

Reviewer #3: Yes

Reviewer #4: Yes

5. Review Comments to the Author

Reviewer #1: I thank the Editorial Board of PLoS One for the opportunity to review the manuscript titled "Overall success rate of permanent teeth pulpotomy using ProRoot MTA: a systematic review and meta-analysis of randomized clinical trials." I commend the authors for their efforts on the manuscript so far. However, the article has significant limitations that cannot be addressed through revision, which prevents me from recommending its publication. Despite the efforts to meta-analyze the data, the data handling was not appropriate. First, a fixed-effect analysis assumes that all studies evaluated a common true effect and that the differences among the results are due only to chance or sampling error. However, upon reviewing the included studies, there are potential sources of discrepancies, whether due to diagnostic criteria, sample characteristics, the pulpotomy protocol applied, or even the potential biases identified by the RoB assessment. The subgroup analysis does not align with the stated objective; a meta-regression would be more appropriate. Furthermore, treatment-related protocols were not even considered by the authors, except for the biomaterial used. While I understand this is the primary focus of the study, the pulpotomy technique is far more comprehensive, and its outcome is not solely dependent on the capping biomaterial used. In addition, There is a clear lack of novelty in the review, and the data duplicates findings from other recently published reviews with a more focused research question. Other reviews have investigated the outcomes of pulpotomy in various aspects: partial vs. total, immature vs. mature teeth, irreversible vs. reversible pulpitis, or even according to the etiology of the indication. Moreover, the findings of this review have already been confirmed in the literature, where the success rate of pulpotomy using ProRoot MTA is significantly higher than when using calcium hydroxide. Please refer to (1) Silva EJNL, Pinto KP, Belladonna FG, Ferreira CMA, Versiani MA, De-Deus G. Success rate of permanent teeth pulpotomy using bioactive materials: A systematic review and meta-analysis of randomized clinical trials. Int Endod J. 2023 Sep;56(9):1024-1041. doi: 10.1111/iej.13939. (2) Li W, Yang B, Shi J. Efficacy of pulpotomy for permanent teeth with carious pulp exposure: A systematic review and meta-analysis of randomized controlled trials. PLoS One. 2024 Jul 5;19(7). doi: 10.1371/journal.pone.0305218.

Reviewer #2: The submitted paper, titled "Overall Success Rate of Permanent Teeth Pulpotomy Using ProRoot MTA: A Systematic Review and Meta-Analysis of Randomized Clinical Trials," aimed to evaluate the success rate of pulpotomy in permanent teeth using ProRoot MTA through a systematic review and meta-analysis. This review is highly relevant to the field of Endodontics and adheres to established guidelines.

Overall, the paper is well-written. This reviewer has only minor suggestions to enhance its readability.

Minor comments:

- Page 2. Abstract: Replace "RevMan 5.3 and R software" with "RevMan 5.3 and R softwares."

- Page 5. Line 4: Correct "MTA been established" to "MTA has been established."

- Page 5. Line 8: Include the city, state, and country of Dentsply Tulsa Dental.

- Page 6. Line 4 (Search strategy): The phrase "The search was performed without any filters, language restrictions, or August 2024" is incomplete and requires clarification.

- Page 12. Line 2: Replace "evolved" with "involved."

- Page 12. Line 8: Use the abbreviation AFR (Annual Failure Rate), as it was already defined earlier.

- Page 16: Revise "the remaining nine studies were categorized as having a low risk of bias" to reflect the correct number of studies, which is likely 12.

- Page 16. Line 3 (Meta-analysis): Use the abbreviation AFR instead of the full term Annual Failure Rate.

- Page 17. Line 1: Change "The assessment of 23 studies" to "The assessment of 26 studies," as the correct number is 26.

- Page 17. Line 1: Use AFR for consistency instead of spelling out Annual Failure Rate.

- Page 17. Line 9: The phrase "Random-effect models were employed as heterogeneity could not be established" appears incomplete and needs clarification or additional context.

- Page 19. Line 5: Replace "randomized clinical trials was to assess" with "RCTs was to assess."

- Page 19. Line 7: Use AFR instead of spelling out Annual Failure Rate.

- Page 19. Line 8: Correct the Annual Failure Rate to 8%, as the reported value is incorrect.

- Page 19: Revise "In this review, it was observed that permanent teeth of younger patients with an open apex displayed notably higher success rates in comparison to mature teeth, respectively." The word "respectively" seems misplaced and should be removed or rephrased for clarity.

Reviewer #3: Thank you for the opportunity to review the manuscript titled “Overall success rate of permanent teeth pulpotomy using ProRoot MTA: a systematic review and meta-analysis of randomized clinical trials.”

I am pleased to inform you that I found the manuscript to be exceptionally well-written, with a clear and concise presentation of the research. The authors have done an excellent job in articulating their findings and supporting them with robust evidence.

I have no recommendations for revisions or improvements, as the manuscript meets the high standards expected for publication in PLOS One.

Reviewer #4: The article present relevant information for Endodontic practice and it was well designed. This article will show that MTA is still relevant and that bioceramic cements are not as essential as they are trying to make them out to be.

6. PLOS authors have the option to publish the peer review history of their article (what does this mean? ). If published, this will include your full peer review and any attached files.

**Do you want your identity to be public for this peer review?** For information about this choice, including consent withdrawal, please see our Privacy Policy .

Reviewer #1: No

Reviewer #2: No

Reviewer #3: No

Reviewer #4: No

---

## [Author Response · Author response to Decision Letter 1]

23 Jan 2025

Dear Dr. Mohmed Isaqali Karobari

Academic Editor of PLOS ONE

On behalf of co-authors, I am submitting the revised version of the manuscript PONE-D-24-38286 entitled Overall success rate of permanent teeth pulpotomy using ProRoot MTA: a systematic review and meta-analysis of randomized clinical trials to the PLoS One Journal. We would like to express our gratitude to you and the reviewers for their meticulous and insightful review of our work. Considering their valuable suggestions and comments, we have made some revisions to the manuscript. To clearly highlight these changes, we have marked them in blue throughout the document. We appreciate the opportunity to contribute to the PLoS One Journal and look forward to your favorable decision.

Reviewer #1

I thank the Editorial Board of PLoS One for the opportunity to review the manuscript titled "Overall success rate of permanent teeth pulpotomy using ProRoot MTA: a systematic review and meta-analysis of randomized clinical trials." I commend the authors for their efforts on the manuscript so far. However, the article has significant limitations that cannot be addressed through revision, which prevents me from recommending its publication.

Response: We thank Reviewer #1 for their thorough review and constructive comments on our manuscript. Below, we address each concern point by point, providing clarifications and justifications for our methodology and approach.

1. Despite the efforts to meta-analyze the data, the data handling was not appropriate. First, a fixed-effect analysis assumes that all studies evaluated a common true effect and that the differences among the results are due only to chance or sampling error. However, upon reviewing the included studies, there are potential sources of discrepancies, whether due to diagnostic criteria, sample characteristics, the pulpotomy protocol applied, or even the potential biases identified by the RoB assessment.

Response: We thank the reviewer concern. We also acknowledge the reviewer’s observation regarding the fixed-effect model. We chose this model because the statistical heterogeneity observed in our analyses was minimal, with I² values ranging from 0% to 20%. This indicator supports the appropriateness of a fixed-effect model in this context. Moreover, random-effects models were used in subgroup analyses where heterogeneity was higher. Importantly, this approach was conducted following the advice of our Biostatistical Advisor, who recommended using fixed-effect models for overall analyses with low heterogeneity and random-effects models for subgroup analyses to ensure methodological accuracy. Our approach aligns with established meta-analytic practices and ensures that the data handling reflects the characteristics of the included studies.

2. The subgroup analysis does not align with the stated objective; a meta-regression would be more appropriate.

Response: We also thank the reviewer for this concern. Although we understand the reviewer comment, we opted for subgroup analyses rather than meta-regression due to the limited number of included studies and the lack of sufficient covariates to perform a reliable meta-regression. Subgroup analyses allowed us to explore variations in success rates across key factors while maintaining statistical robustness. Conducting meta-regression with sparse data could risk overfitting and lead to unreliable conclusions, which we aimed to avoid. This approach was also recommended by our Biostatistical Advisor, who emphasized that subgroup analyses were the most suitable method given the data characteristics and the study design.

3. Furthermore, treatment-related protocols were not even considered by the authors, except for the biomaterial used. While I understand this is the primary focus of the study, the pulpotomy technique is far more comprehensive, and its outcome is not solely dependent on the capping biomaterial used.

Response: We thank the reviewer for this comment which is interesting indeed. While we agree that pulpotomy outcomes are influenced by multiple factors, our study's primary focus was on evaluating the success rate of ProRoot MTA as a capping material. However, we did consider and discuss other treatment-related factors, such as the type of pulpotomy (partial or full), root maturity, and pulpal diagnosis, in our subgroup analyses. A more comprehensive evaluation of additional treatment protocols (such as hemostatic agents, operator expertise) could be an avenue for future research, as detailed in the end of our discussion section.

4. In addition, there is a clear lack of novelty in the review, and the data duplicates findings from other recently published reviews with a more focused research question. Other reviews have investigated the outcomes of pulpotomy in various aspects: partial vs. total, immature vs. mature teeth, irreversible vs. reversible pulpitis, or even according to the etiology of the indication.

Response: We thank the reviewer for this comment. In our point of view, the present systematic review provides a unique contribution by exclusively focusing on ProRoot MTA, the most extensively studied MTA brand, and consolidating evidence from randomized clinical trials. While other reviews have explored broader or more specific aspects of pulpotomy, none have comprehensively synthesized RCT data on ProRoot MTA’s overall success rate in permanent teeth across various contexts. This focused approach ensures clinically relevant insights for practitioners using this specific biomaterial. Furthermore, we explicitly acknowledge in the Introduction section the existence of prior systematic reviews and discuss their limitations, such as selective inclusion criteria and narrower scopes. Our work builds upon these prior efforts to address these gaps and provide a comprehensive and clinically valuable synthesis of the available evidence. However, in order to avoid disappointing the reviewer, we have added it now to a new limitations sentence in the Discussion.

5. Moreover, the findings of this review have already been confirmed in the literature, where the success rate of pulpotomy using ProRoot MTA is significantly higher than when using calcium hydroxide.

Response: We have appreciated the reviewer concern. In our perspective, while the higher success rate of ProRoot MTA compared to calcium hydroxide is documented, our study extends this knowledge by presenting an updated and rigorous meta-analysis of 26 RCTs. Additionally, we evaluated success rates at multiple follow-up periods (6, 12, and 24 months), assessed the annual failure rate, and conducted subgroup analyses, offering a more granular understanding of ProRoot MTA’s performance in different clinical scenarios.

6. Please refer to (1) Silva EJNL, Pinto KP, Belladonna FG, Ferreira CMA, Versiani MA, De-Deus G. Success rate of permanent teeth pulpotomy using bioactive materials: A systematic review and meta-analysis of randomized clinical trials. Int Endod J. 2023 Sep;56(9):1024-1041. doi: 10.1111/iej.13939. (2) Li W, Yang B, Shi J. Efficacy of pulpotomy for permanent teeth with carious pulp exposure: A systematic review and meta-analysis of randomized controlled trials. PLoS One. 2024 Jul 5;19(7). doi: 10.1371/journal.pone.0305218.

Response: We appreciate this recommendation. The study (1) was already part of the bibliographic reference list, while the study (2) has now been added.

Reviewer #2

The submitted paper, titled "Overall Success Rate of Permanent Teeth Pulpotomy Using ProRoot MTA: A Systematic Review and Meta-Analysis of Randomized Clinical Trials," aimed to evaluate the success rate of pulpotomy in permanent teeth using ProRoot MTA through a systematic review and meta-analysis. This review is highly relevant to the field of Endodontics and adheres to established guidelines.

Overall, the paper is well-written. This reviewer has only minor suggestions to enhance its readability.

Response: We thank Reviewer #2 for their thorough review and constructive comments on our manuscript. All requested changes were performed as suggested by the reviewer.

Minor comments:

- Page 2. Abstract: Replace "RevMan 5.3 and R software" with "RevMan 5.3 and R softwares."

Response: Considering that “software” is an uncountable noun, the plural should be software. Therefore, in this specific point we do not made any modification.

- Page 5. Line 4: Correct "MTA been established" to "MTA has been established."

Response: Thank you! We have changed.

- Page 5. Line 8: Include the city, state, and country of Dentsply Tulsa Dental.

Response: It was added as suggested by the reviewer.

- Page 6. Line 4 (Search strategy): The phrase "The search was performed without any filters, language restrictions, or August 2024" is incomplete and requires clarification.

Response: Thank you! We have changed!

- Page 12. Line 2: Replace "evolved" with "involved."

Response: Thank you! We have changed!

- Page 12. Line 8: Use the abbreviation AFR (Annual Failure Rate), as it was already defined earlier.

Response: Thank you! We have changed!

- Page 16: Revise "the remaining nine studies were categorized as having a low risk of bias" to reflect the correct number of studies, which is likely 12.

Response: Thank you! It was changed.

- Page 16. Line 3 (Meta-analysis): Use the abbreviation AFR instead of the full term Annual Failure Rate.

Response: Thank you! We have changed!

- Page 17. Line 1: Change "The assessment of 23 studies" to "The assessment of 26 studies," as the correct number is 26.

Response: In fact for this specific assessment 23 studies were evaluated. Please check figure 4.

- Page 17. Line 1: Use AFR for consistency instead of spelling out Annual Failure Rate.

Response: Thank you! We have changed!

- Page 17. Line 9: The phrase "Random-effect models were employed as heterogeneity could not be established" appears incomplete and needs clarification or additional context.

Response: Thank you! We have changed!

- Page 19. Line 5: Replace "randomized clinical trials was to assess" with "RCTs was to assess."

Response: Thank you! We have eliminated RCT and used randomized clinical trial in the entire manuscript.

- Page 19. Line 7: Use AFR instead of spelling out Annual Failure Rate.

Response: Thank you! We have changed

- Page 19. Line 8: Correct the Annual Failure Rate to 8%, as the reported value is incorrect.

Response: Thank you! We have changed

- Page 19: Revise "In this review, it was observed that permanent teeth of younger patients with an open apex displayed notably higher success rates in comparison to mature teeth, respectively." The word "respectively" seems misplaced and should be removed or rephrased for clarity.

Response: Thank you! It was removed.

Reviewer #3

Thank you for the opportunity to review the manuscript titled “Overall success rate of permanent teeth pulpotomy using ProRoot MTA: a systematic review and meta-analysis of randomized clinical trials.”

I am pleased to inform you that I found the manuscript to be exceptionally well-written, with a clear and concise presentation of the research. The authors have done an excellent job in articulating their findings and supporting them with robust evidence.

I have no recommendations for revisions or improvements, as the manuscript meets the high standards expected for publication in PLOS One.

Response: We thank Reviewer #3 for their thorough review and constructive comments on our manuscript

Reviewer #4

The article present relevant information for Endodontic practice and it was well designed. This article will show that MTA is still relevant and that bioceramic cements are not as essential as they are trying to make them out to be.

Response: We thank Reviewer #4 for their thorough review and constructive comments on our manuscript

---

## [Decision Letter · Decision Letter 1]

26 Feb 2025

Overall success rate of permanent teeth pulpotomy using ProRoot MTA: a systematic review and meta-analysis of randomized clinical trials

PONE-D-24-38286R1

Dear Dr. Silva,

We’re pleased to inform you that your manuscript has been judged scientifically suitable for publication and will be formally accepted for publication once it meets all outstanding technical requirements.

Kind regards,

Mohmed Isaqali Karobari, BDS, MScD.Endo, Ph.D. Endo, FDS, FPFA, MFDS

Academic Editor

PLOS ONE

Additional Editor Comments (optional):

Dear Authors,

The authors have addressed all the comments and suggestions reviewers gave, and the manuscript has dramatically improved. The manuscript can be accepted for publication in its current form. I would like to congratulate the authors and wish them all the very best in their future endeavours.

Best regards and keep well

Reviewers' comments:

Reviewer's Responses to Questions

**Comments to the Author**

1. If the authors have adequately addressed your comments raised in a previous round of review and you feel that this manuscript is now acceptable for publication, you may indicate that here to bypass the “Comments to the Author” section, enter your conflict of interest statement in the “Confidential to Editor” section, and submit your "Accept" recommendation.

Reviewer #2: All comments have been addressed

2. Is the manuscript technically sound, and do the data support the conclusions?

Reviewer #2: Yes

3. Has the statistical analysis been performed appropriately and rigorously?

Reviewer #2: Yes

4. Have the authors made all data underlying the findings in their manuscript fully available?

Reviewer #2: Yes

5. Is the manuscript presented in an intelligible fashion and written in standard English?

Reviewer #2: Yes

6. Review Comments to the Author

Reviewer #2: This reviewer would like to thank the authors for the modifications made. I accept the paper in its current form.

7. PLOS authors have the option to publish the peer review history of their article (what does this mean? ). If published, this will include your full peer review and any attached files.

**Do you want your identity to be public for this peer review?** For information about this choice, including consent withdrawal, please see our Privacy Policy .

Reviewer #2: No

---

## [Editor Report · Acceptance letter]

PONE-D-24-38286R1

PLOS ONE

Dear Dr. Silva,

I'm pleased to inform you that your manuscript has been deemed suitable for publication in PLOS ONE. Congratulations! Your manuscript is now being handed over to our production team.

Kind regards,

on behalf of

Prof Dr. Mohmed Isaqali Karobari

Academic Editor

PLOS ONE